# Streamlined asymmetric α-difunctionalization of ynones

Siyu Peng[1], Zhaofeng Wang [1], Linxing Zhang[1], Xinhao Zhang[1] & Yong Huang[1]

Ynones are a unique class of structural motifs that show remarkable chemical versatility. Chiral ynones, particularly those possessing an α-stereogenic center, are highly attractive templates for structural diversification. So far, only very limited examples have been reported for asymmetric α-functionalization of ynones. Asymmetric double α-functionalization of ynones remains elusive. Here we describe a streamlined strategy for asymmetric α-difunctionalization of ynones. We developed a gold-catalyzed multicomponent condensation reaction from a simple ynone, an amine, and an electrophilic alkynylating reagent to generate a 1,2-dialkynyl enamine, a key stable and isolable intermediate. This intermediate can undergo asymmetric fluorination catalyzed by a chiral phosphoric acid derivative. Chiral ynones with an α-quaternary carbon and containing a fluorine and an alkyne can be synthesized in high yield and high ee. The synthetic utility of this method is demonstrated by the synthesis of enantioenriched tri(hetero)arylmethyl fluorides.

[1] Key Laboratory of Chemical Genomics, Peking University, Shenzhen Graduate School, 518055 Shenzhen, China. Siyu Peng and Zhaofeng Wang contributed equally to this work. Correspondence and requests for materials should be addressed to Y.H. (email: huangyong@pkusz.edu.cn)

Ynones are abundant structural motifs and versatile synthetic intermediates that are useful in the synthesis of natural products and other complex molecular structures[1–3]. The unique property and reactivity of alkynes, coupled with the chemical versatility of carbonyl compounds, makes ynones an indispensable template for rapid structural proliferation and diversification[4–7]. Consequently, ynones possessing an α-chiral center are highly attractive chiral synthetic building blocks[8,9]. Although numerous synthetic approaches to ynones have been reported, the synthesis of chiral ynones remains a significant challenge due to multiple chemically reactive sites that interfere with common catalysts. Direct α-functionalization of carbonyl compounds is the most straightforward strategy for the preparation of α-chiral ketones.

Despite considerable advances in the field[10–21] since the first intermolecular asymmetric aldol reaction was reported by List and Barbas[22], ynones, as a substrate for both enamine and enolate chemistry, have been underexplored. The presence of carbon–carbon triple bond conjugated with a carbonyl creates a number of challenges. The π-basicity of an alkyne makes it an excellent ligand for transition metals[23–27]. The acetylene group is linear and small, making enamine/enolate E/Z geometry and facial discrimination difficult. In addition, ynones are highly active Michael acceptors that react with various nucleophiles[28]. For these reasons, there are very few reports describing synthetic methods producing α-chiral ynones.

In 2006, Gouverneur et al. reported the first enantioselective aldol reaction of ynones using catalysis by secondary amines (Fig. 1a)[29]. In 2009, Trost et al. reported a palladium-catalyzed decarboxylative asymmetric allylic alkylation of allyl enol carbonates[30]. Trost's report had one example involving an ynone. Recently, the same group reported the first catalytic Mannich-type reaction of ynones and N-Boc imines, producing β-amino ynones with excellent chemoselectivity, diastereoselectivity, and enantioselectivity (Fig. 1b)[31]. In 2009, Davies et al.

reported an enantioselective cyclopropanation of donor/acceptor carbenoids using α-diazoketones[32]. Several ynone substrates were covered (Fig. 1c). To the best of our knowledge, this two-step transformation is the only report of the synthesis of ynones with an α-quaternary chiral center, and to date, α-difunctionalization of ynones, by introduction, asymmetrically, of two distinct functional groups at the same α-carbon center, remains an elusive goal. Here we describe our recent progress on asymmetric α-alkynylation and α-fluorination of ynones by a streamlined reaction sequence involving a diynenamine intermediate. Our strategy involves a key diynenamine intermediate that can be accessed by direct α-alkynylation of ynones. The subsequent asymmetric α-fluorination is accomplished using a chiral phosphoric acid and a chiral diynenamine. Interestingly, the chirality of the newly formed α-quaternary carbon center is controlled by the absolute stereochemistry of the chiral diynenamine. This streamlined synthesis gives rise to an interesting class of chiral centers containing four orthogonal functional groups: aryl, alkyne, ynone, and fluorine. Subsequent synthetic manipulations of the products lead to chiral tri(hetero)arylmethyl fluorides.

## Results

**Synthesis of diynenamines.** Our group has a long-held interest in studies of highly functionalized enamine species and their application to novel chemical transformations[33–38]. In 2013, we reported direct α-alkynylation of aldehydes, using gold/amine synergistic catalysis, to obtain ynenamine intermediates that provide access to trisubstituted and tetrasubstituted allenes[33]. Subsequently, we synthesized ynedienamines, which show unique chemical reactivity toward a broad spectrum of electrophiles and nucleophiles[34,35]. In addition, we reported ester-substituted enamines that can be used in an enantioselective Povarov reaction and α-iodoenamine useful in direct α-cyclopropanation of aldehydes[36,37]. To access α-difunctionalized ynones, we proposed

**Fig. 1** Asymmetric α-functionalization of ynones. **a** Enantioselective aldol reaction of ynones; **b** enantioselective Mannich reaction of ynones; **c** generation of ynones with an α-quaternary chiral center using cyclopropanation; **d** asymmetric α-alkynylation–α-fluorination of ynones

**Table 1 Condition survey for the synthesis of diynenamines[a,b]**

| Entry | T (°C) | Ligand | Solvent | Yield (%) |
|-------|--------|--------|---------|-----------|
| 1 | rt | — | Toluene | 55 |
| 2 | rt | — | Et₂O | 68 |
| 3 | rt | Py | Et₂O | 65 |
| 4 | rt | 2,2′-Bpy | Et₂O | 75 |
| 5 | rt | 1,10-Phen | Et₂O | 70 |
| 6 | rt | DAFO | Et₂O | 72 |
| 7 | 50 | 2,2′-Bpy | Et₂O | 88 |

Py pyridine, 2,2′-Bpy 2,2′-bipyridine, DAFO 4,5-diazofluoren-9-one
[a] Reactions were performed using **1a** (0.1 mmol), TIPS-EBX (0.12 mmol), AuCl (0.01 mmol), pyrrolidine (0.12 mmol), and ligand (0.02 mmol) in a solvent (2.0 mL) under Ar for 10 h
[b] Isolated yield

a fully conjugated diynenamine intermediate that can be functionalized in a streamlined fashion. We anticipated that, in the presence of both a secondary amine and a gold catalyst, an ynone might react with electrophilic alkynylating reagents such as 1-[(triisopropylsilyl)-ethynyl]-1,2-benziodoxol-3(1 H)-one (TIPS-EBX)[33,39–46] to give a diynenamine species, which, upon hydrolysis, yields α-alkynyl ynones. This step need not be asymmetric as the second α-functionalization will go through the same achiral diynenamine intermediate. The stereochemistry of resulting quaternary α-carbon center can be controlled during the second functionalization step.

We began our study using 1-phenyl-4-(triisopropylsilyl)but-3-yn-2-one (**1a**), which can be easily accessed by our ynenamine chemistry[38]. We found that, in the presence of 1.2 equivalents of pyrrolidine, 1.2 equivalents of TIPS-EBX and 10 mol% AuCl, the diynenamine (**3a**) was isolated in 55% yield with >20:1 Z/E ratio (Table 1, entry 1). The highly conjugated π system probably contributes to the stability of this unique enamine species[33]. The double bond was assigned a Z-configuration with the two alkynes in *trans* orientation, which is supported by lack of Overhauser effects in the NOE studies (See Supplementary Figure 118). The diynenamine (**3a**) is stable during silica gel column chromatography and displays a bright yellow color as a result of the extended conjugation. Reaction variables were examined systematically. Diethyl ether was found to be the optimum solvent for this transformation. Although the reaction can proceed without a ligand (Table 1, entries 1 and 2), the use of 20 mol% 2,2′-bipyridine led to an improved yield (Table 1, entries 4–6). The involvement of bipyridine ligands makes the gold species more electron deficient, thus leading to a better reactivity in the activation of the EBX reagent, consistent with our previous study[33]. Finally, raising the reaction temperature to 50 °C afforded **3a** in 88% isolated yield (Table 1, entry 7).

**Reaction scope of ynones and amines**. With the optimized conditions in hand, we explored the substrate scope for both the ynone and the amine. Substituents with various electronic characteristics on the phenyl group are well tolerated (Table 2,

products **3a–3k**). Ynones containing other aryls and heteroaryls, naphthyl and furyl, for example, are also suitable substrates for this reaction (Table 2, products **3l–3n**). For TIPS- protected ynones, good-to-excellent Z/E ratios were obtained. When the terminal TIPS group of the ynone was replaced with the smaller silyl (Table 2, products **3q** and **3r**), alkyl (Table 2, products **3s** and **3t**), aryl or heteroaryl groups (Table 2, products **3u–3z**), good conversion with moderate Z/E selectivity was observed. The lower Z/E ratio is thought to be a result of attenuated steric repulsion between the two alkyne groups. Aryl ketones do not interfere with the diynenamine formation (Table 2, product **3y**) and common electrophiles such as alkyl chlorides are tolerated. Various amines were also examined. Consistent with our previous observation[34], only cyclic amines are effective for this transformation, giving products **3aa–3ad**. This limitation is inconsequential, however, as the amine is hydrolyzed in the second α-functionalization step.

**Asymmetric α-fluorination using diynenamines**. We next explored asymmetric α-fluorination using the diynenamines **3** (Table 3). Substrate **3a** reacts smoothly with N-fluorobenzenesulfonimide (NFSI) in chloroform at room temperature. The desired α-alkynyl-α-fluoroynone (**4a**) was obtained in 1 h as a racemic mixture in quantitative yield. We were inspired by Toste's pioneering work on enantioselective α-fluorination using a chiral phosphoric acid (CPA) as a phase-transfer catalyst, together with a poorly soluble fluorinating reagent[47–60]. The combination of CPA (**6a**) and NFSI led to racemic **4a**. Observation of this rapid, uncatalyzed fluorination with NFSI prompted us to search for other fluorinating agents. N-fluoropyridinium salts failed to yield any product, but in the presence of CPA (**R-6a**) and Selectfluor® (1-chloromethyl-4-fluoro-1,4-diazoniabicyclo[2.2.2]octane bis(tetrafluoroborate) in chloroform, product **4a** was obtained in excellent yield and 11% enantiomeric excess (ee). The low ee in this case could be attributed to the high solubility of Selectfluor® in chloroform. Toste showed that a poor solvent for Selectfluor®, toluene or hexane, for example, is critical to ensure that all F⁺ species in solution are bonded to the CPA anion, but unfortunately, switching to toluene only resulted in 12% ee of **4a**.

**Table 2 Substrate scope of ynones and amines[a]**

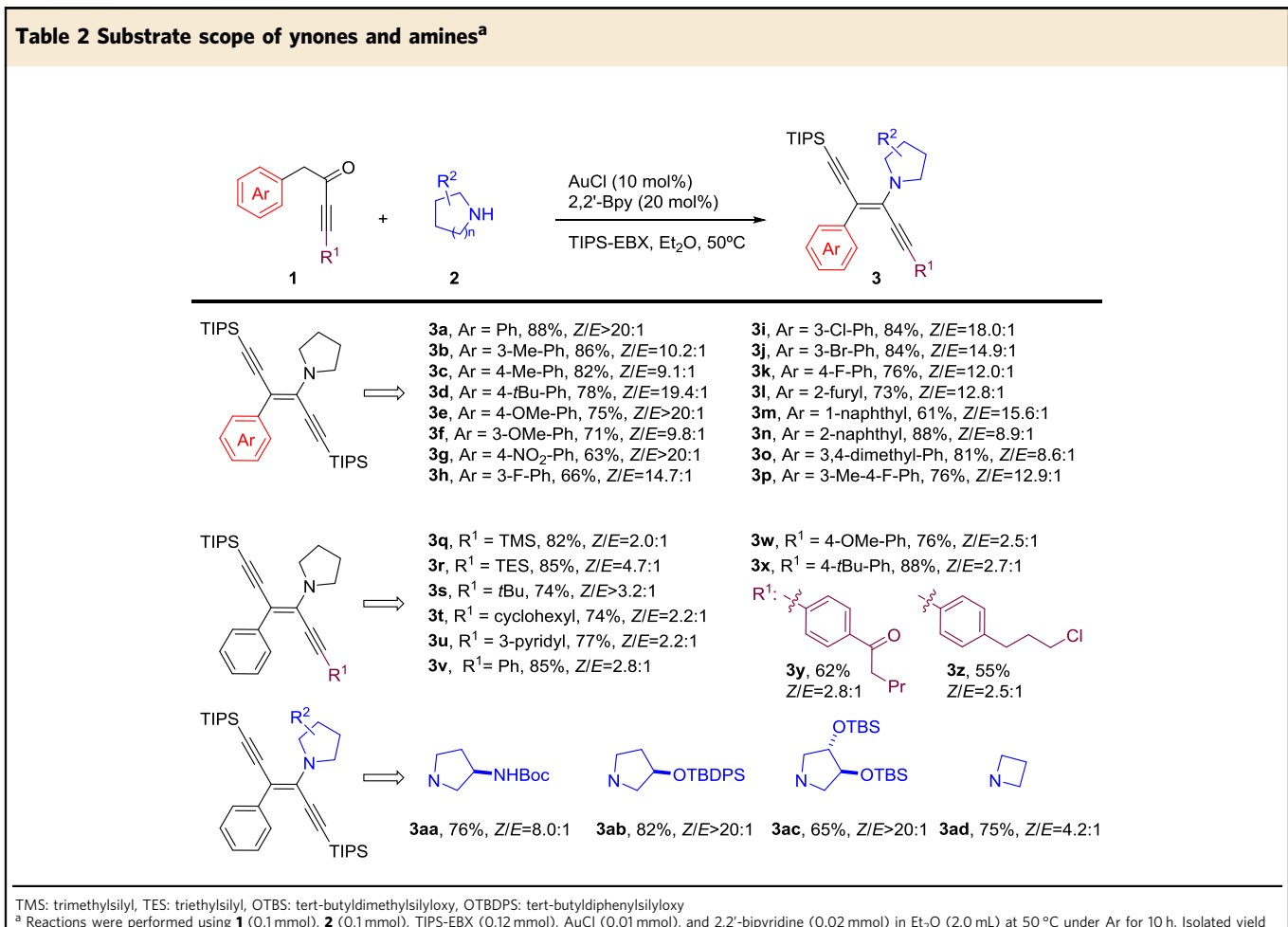

TMS: trimethylsilyl, TES: triethylsilyl, OTBS: tert-butyldimethylsilyloxy, OTBDPS: tert-butyldiphenylsilyloxy
[a] Reactions were performed using **1** (0.1 mmol), **2** (0.1 mmol), TIPS-EBX (0.12 mmol), AuCl (0.01 mmol), and 2,2′-bipyridine (0.02 mmol) in Et₂O (2.0 mL) at 50 °C under Ar for 10 h. Isolated yield

Considering the small size of the alkyne group, it is not surprising that the asymmetric fluorination of the diynenamine (**3a**) would be challenging and we decided to use a chiral diynenamine to enhance the stereoselectivity of this reaction. Interestingly, the diynenamine **R-3ab** reacted with NSFI at 0 °C, giving product **4a** with 36% ee. Introduction of an (R)-3-OTBDPS group resulted in a faster reaction. The use of **S-3ab** and CPA **R-6a** led to 20 and 39% ee in chloroform and toluene, respectively. The selectivity of the α-fluorination steadily increases with the size of the 3,3′-substituents of the CPA[61–65]. The α-fluorination occurred in 88% yield with 91% ee using **R-3ab** and CPA **S-6e** at −10 °C (Table 3, entry 13). In comparison, the mismatched combination of **R−3ab** and **R-6e** resulted in 60% ee under identical conditions. The absolute configuration of the product is controlled by the chiral amino group, rather than by the CPA, whose role is enhancement of the facial discrimination imposed by the chiral amine. The reaction became sluggish in toluene when cooled to −30 °C, but both high yield and high ee were obtained when using *m*-xylene as the solvent (Table 3, entry 16).

**Substrate scope of the α-difunctionalization sequence**. The reaction scope of the double functionalization sequence, α-alkynylation–α-fluorination was surveyed. The ynone was subjected to the diynenamine formation under the standard conditions shown in Table 2. The crude mixture was passed through a short column of Florisil®. The resulting crude diynenamine was then dissolved in *m*-xylene and fluorinated according to the optimized conditions identified in Table 3. Various alkyl groups on the

aromatic ring are tolerated (Table 4, products **4b** and **4d**) and good overall yield and ee were obtained. The selectivity of the reaction is not affected by either electron-withdrawing or electron-donating functionalities on the phenyl ring. Ynones bearing a naphthyl or a polysubstituted aryl ring react well (Table 4j–l). Although a stoichiometric amount of the chiral amine is used, this chiral amine is relatively cheap and can be recovered and recycled (77% recovery yield). The absolute stereochemistry of **4a** was assigned the *R* configuration by the X-ray structure of its derivative (Table 4, compound 7). The ee of the product can be correlated to the *Z/E* ratio of the diynenamine intermediates. As mentioned earlier, modest *Z/E* selectivity was observed when R¹ is an alkyl or aryl group and a moderate ee was obtained for these substrates (Table 4, **4m** and **4n**).

**Sythesis of tri(hetero)arylmethyl fluorides**. The chirality of product **4** is intriguing. This chiral center contains four distinct functional groups: aryl, alkyne, ynone, and fluorine, and offers access to a new chemical space. To showcase the synthetic versatility of this product, we prepared a series of chiral triarylmethyl fluorides. Although there are a number of synthetic methods leading to the pharmacologically relevant chiral triarylmethanes[66,67], the corresponding chiral triarylmethyl fluorides are not readily accessible. The orthogonal reactivities of the alkyne and the ynone functionalities allow us to construct heteroaryl groups sequentially. The ynone moiety reacts with benzamidine hydrochloride to form a substituted pyrimidine (Fig. 2, product **8**) in 90% yield[68]. The remaining alkynyl group, after

**Table 3 Asymmetric α-fluorination of diynenamines[a,b]**

R = H, **3a**
R = R-OTBDPS, **R-3ab**
R = S-OTBDPS, **S-3ab**

**6a**    **6b**    **6c**, R= OH; **6d**, R= NHTf    **6e**

| Entry | Sub | F*[b] | CPA | solvent | T (°C) | Yield (%) | ee (%) |
|---|---|---|---|---|---|---|---|
| 1 | 3a | 5a | — | CHCl$_3$ | rt | quant. | 0 |
| 2 | 3a | 5a | R-6a | CHCl$_3$ | rt | quant. | 0 |
| 3 | 3a | 5b | R-6a | CHCl$_3$ | rt | NR | ND |
| 4 | 3a | 5c | R-6a | CHCl$_3$ | rt | 95 | 11 |
| 5 | 3a | 5c | R-6a | Toluene | rt | 89 | 12 |
| 6 | R-3ab | 5a | — | DCM | 0 | quant. | −36 |
| 7 | S-3ab | 5c | R-6a | CHCl$_3$ | rt | 90 | 20 |
| 8 | S-3ab | 5c | R-6a | Toluene | rt | 90 | 39 |
| 9 | S-3ab | 5c | R-6b | Toluene | rt | 95 | 41 |
| 10 | S-3ab | 5c | R-6c | Toluene | rt | 90 | 72 |
| 11 | S-3ab | 5c | R-6d | Toluene | rt | 90 | 82 |
| 12 | S-3ab | 5c | R-6e | Toluene | rt | 90 | 80 |
| 13 | R-3ab | 5c | S-6e | Toluene | −10 | 88 | −91 |
| 14 | R-3ab | 5c | R-6e | Toluene | −10 | 89 | −60 |
| 15 | R-3ab | 5c | S-6e | Toluene | −30 | 65 | −91 |
| 16 | R-3ab | 5c | S-6e | m-xylene | −30 | 93 | −92 |
| 17 | 3a | 5c | S-6e | m-xylene | −30 | 93 | −31 |

[a] Reactions were performed using 3 (0.05 mmol), fluorinating reagent 5 (0.06 mmol), CPA (0.005 mmol), and Na$_2$CO$_3$ (0.10 mmol) in a solvent (1.0 mL) for 24–48 h
[b] **5a:** NFSI, **5b:** N-fluoropyridinium tetrafluoroborate, **5c:** Selectfluor®

deprotection, undergoes triazole formation with benzyl azide[69] and isoquinolinone formation upon Rh-catalyzed C-H activation[70] to give triaryl fluoromethane products (**9**, **10**) with complete retention of optical purity.

**Proposed rationale for the asymmetric α-fluorination.** The conformation of the C-N bond in intermediate **R-3ab** is essential to the asymmetric α-fluorination. We carried out a density functional theory (DFT) calculation (B3LYP-D3BJ/6–311 + G(d, p) at 243.15 K, SMD in m-xylene) for the two possible rotamers **R-3ab-R1** and **R-3ab-R2** (Supplementary Note 1). The calculation showed that rotamer **R-3ab-R1** is 1.7 kcal/mol more stable in Gibbs free energy than **R-3ab-R2** (Fig. 3). The re-face of **R-3ab-R1** is exposed as a result of si-face shielding by C5. However, the minimized conformation of **R-3ab-R1** suggests that the phenyl ring might twist out of planarity from the enamine moiety, which compromises the facial bias by C5. As a result, the ee using the amine alone is poor. Based on the experimental results, S-CPA

(**S-6e**) forms a matching pair with the R-amine, which leads to high enantioselectivity. We speculate that better fitting of the R-diynenamine into the S-CPA pocket might be responsible for this result (Fig. 3).

**Discussion**
We have developed a streamlined synthetic protocol for asymmetric α-difunctionalization of ynones. The overall transformation is based on a key diynenamine intermediate, which is prepared by gold-catalyzed multi-component α-alkynylation of ynones. The challenging asymmetrical fluorination of the diynenamine species is accomplished using a combination of a chiral phosphoric acid and a recyclable chiral amine auxiliary. A unique class of compounds with chiral substructures containing four orthogonal functional groups are prepared in high yield and excellent ee. The α-alkynyl–α-fluoro ynone products are further transformed to triarylfluoromethane structures.

**Table 4 Substrate scope of the asymmetric α-alkynylation–α-fluorination**

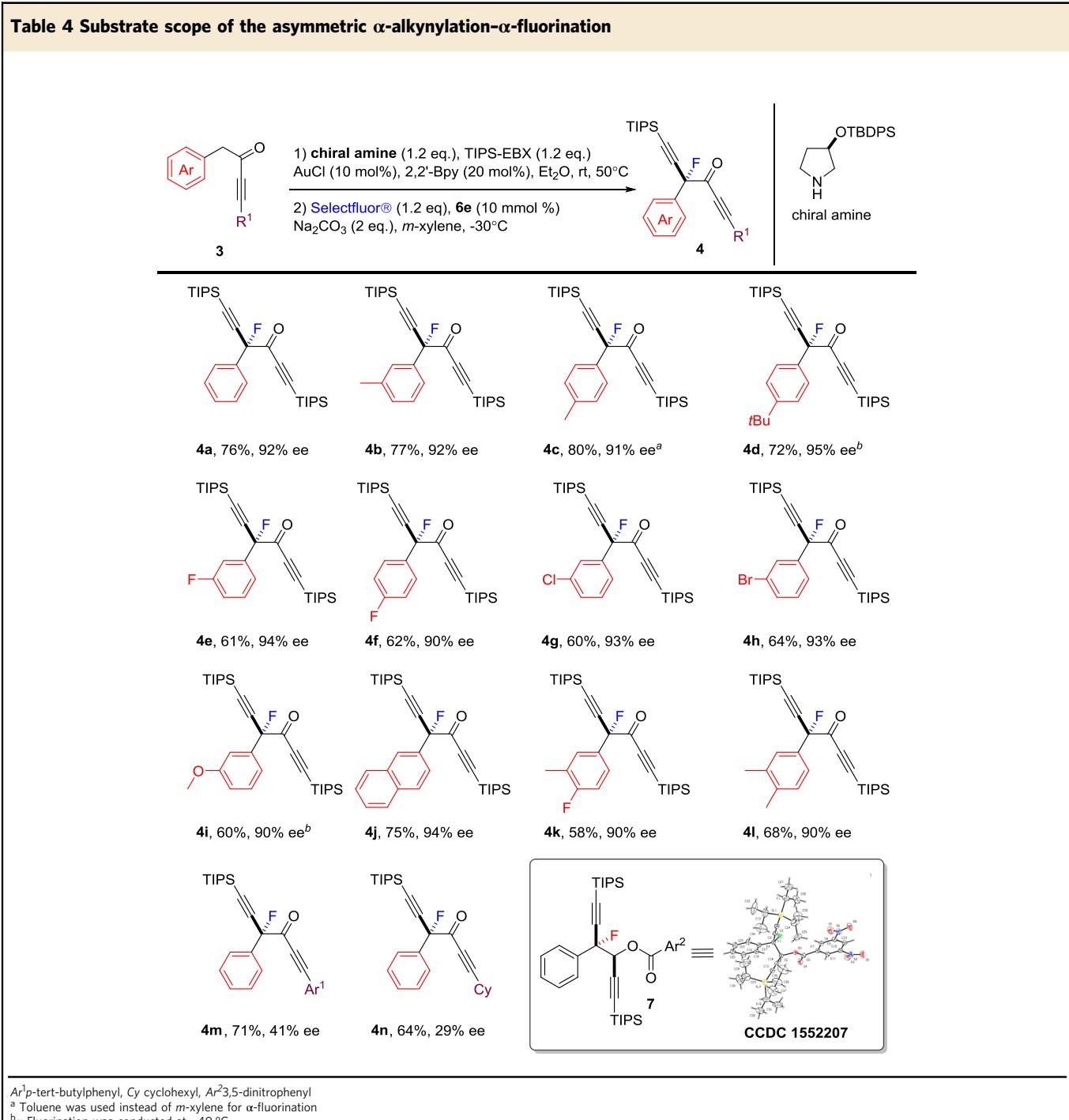

Ar[1] p-tert-butylphenyl, Cy cyclohexyl, Ar[2] 3,5-dinitrophenyl
[a] Toluene was used instead of m-xylene for α-fluorination
[b] α-Fluorination was conducted at −40 °C

## Methods

**General methods and materials**. All solvents were distilled before use. Commercial reagents were used without further purification. HG/T2354-92 silica gel from Qingdao Haiyang Chemical was used for silica gel flash chromatography. Nuclear magnetic resonance (NMR) spectra were taken on Bruker nuclear magnetic resonance spectrometers. Chemical shifts ($\delta$) in ppm are reported; references are according to the chemical shifts of CDCl$_3$ ([1]H 7.26 ppm or [13]C 77.16 ppm). Multiplicities are described as: s = singlet, bs = broad singlet, d = doublet, t = triplet, q = quartet, m = multiplet. Coupling constants are reported in Hz. [13]CNMR spectra were taken with total proton decoupling. Chiral High-performance liquid chromatography (HPLC) was performed on Shimadzu LC-20A systems using Daicel chiral columns. High-resolution mass spectrometric data were recorded on a Bruker Apex IV RTMS instrument. [1]HNMR, [13]CNMR, and [19]FNMR (for fluorine containing compounds) are provided for all products, see Supplementary

Figures 1–119. HPLC spectra are provided for all chiral compounds; see Supplementary Figures 120–151. For ORTEP structure compound **7**, see Supplementary Figure 152. See Supplementary Methods for the characterization data for all compounds. Cartesian coordinates of **R-3ab-R1** and **R-3ab-R2** are listed in Supplementary Notes 2 and 3. See Supplementary Dataset 1 for X-ray CIF file of compounds **7** (Cambridge Crystallographic Data Centre (CCDC) 1552207).

**General procedure for the synthesis of diynenamines**. Ynone **1** (0.10 mmol, 1.0 equiv.), TIPS-EBX (0.12 mmol, 1.2 equiv.), 2,2′-bipyridine (3.1 mg, 0.02 mmol, 0.2 equiv.), and AuCl (2.4 mg, 0.01 mmol, 0.1 equiv.) were dissolved in dry Et$_2$O (2.0 mL, 0.05 M) in an oven-dried 15 mL sealed tube equipped with a magnetic stir bar and a Teflon® stopper. Amine **2** (0.12 mmol, 1.2 equiv.) was added by microsyringe, and the tube was sealed. After stirring for 10 h at 50 °C, the reaction mixture was passed through a short column of Florisil® (adsorbent for

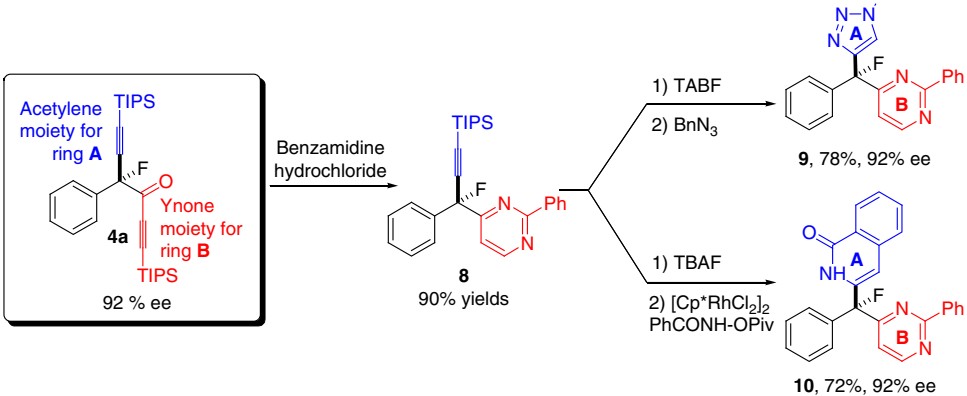

**Fig. 2** Synthetic application of the α-alkynylation–α-fluoro ynone products. Product **8** was obtained by refluxing **4a** with benzamidine hydrochloride (1.2 eq.) in THF/water (7/1) for 16 h. Selective desilylation of **8** was accomplished using tert-butylammonium fluoride (TBAF) in THF at 0 °C. Product **9** was obtained by click chemistry using benzyl azide and CuSO$_4$ and sodium ascorbate in EtOH and water. Product **10** was generated under C-H activation conditions using [Cp*RhCl$_2$]$_2$ and PhCONH-OPiv

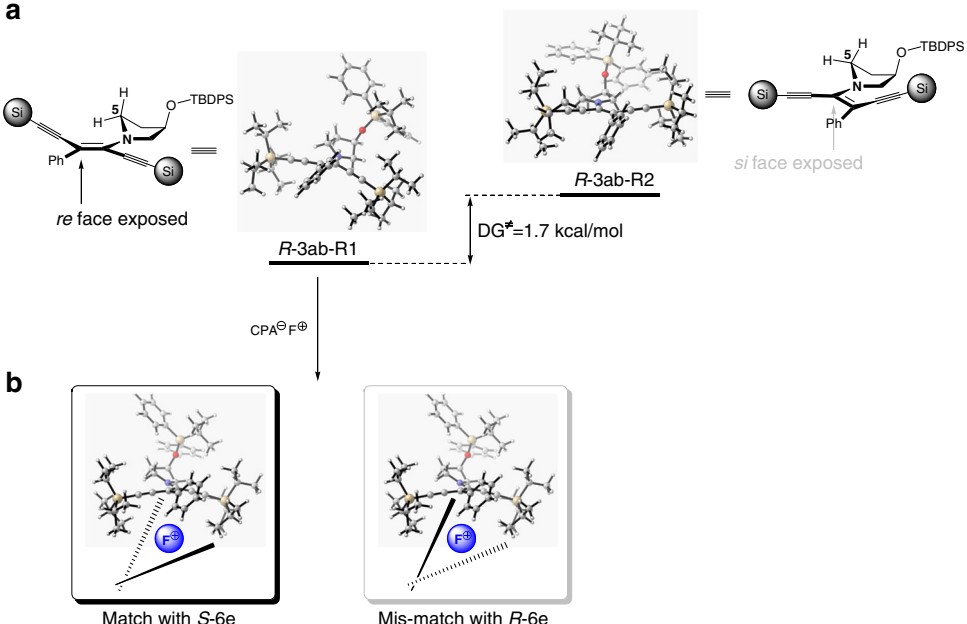

**Fig. 3** Simplified proposal for the stereoselectivity. **a** Minimized conformation of ***R*-3ab-R1** and ***R*-3ab-R2** is represented in ORTEP diagrams; the facial selectivity is likely controlled by the adjacent methylene group (C5); Si = TIPS; **b** the CPA anion is simplified as "chopsticks" for clarity

chromatography florisil, 125–250 μm, Adamas) and washed with Et$_2$O. The eluent was concentrated and purified directly by flash chromatography on a Florisil® adsorbent column to afford the enamine product (**3**).

**Streamlined procedure for the synthesis of chiral ynones**. Ynone **1** (0.10 mmol) **TIPS-EBX** (0.12 mmol), AuCl (0.01 mmol), (*R*)-3-OTBDPS-pyrrolidine (0.12 mmol), and 2,2'-bipyridine (0.02 mmol) were dissolved in Et$_2$O (2.0 mL), and this solution was stirred at 50 °C under Ar for 10–16 h. After the reaction was complete, the diynenamine intermediate was purified by passage through a column of Florisil®. The eluent was concentrated and redissolved in *m*-xylene (0.5 mL). The resulting bright yellow solution was slowly added to a stirred solution of Selectfluor®(**5c**, 0.12 mmol), Na$_2$CO$_3$ (0.20 mmol), and **S-6e** (0.01 mmol) in *m*-xylene (0.5 mL) at −30 °C. The reaction was stirred at −30 °C for 24–48 h until no diynenamine could be detected by thin-layer chromatography. The crude mixture was concentrated and purified by flash chromatography on silica gel column to afford the α,α-difunctionalized ynone (**4**).

**Data availability**. The X-ray crystallographic coordinates for compound **7** reported in this study have been deposited at the CCDC under deposition number

CCDC-1552207. These data can be obtained free of charge from The CCDC via www.ccdc.cam.ac.uk/data_request/cif. The authors declare that other data supporting the findings of this study are available within the paper and its supplementary information files and also are available from the corresponding author upon request.

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

## Acknowledgements

This work is financially supported by the National Natural Science Foundation of China (21572004), Department of Science and Technology of Guangdong Province (2014TX01R111), and the Shenzhen Basic Research Program (JCYJ20160226105602871).

## Author contributions

Y.H. conceived and directed the project. S.P. and Z.W. performed the experiments. L.Z. and X.Z. performed the DFT calculations. Y.H. analyzed the data and wrote the manuscript with input from all authors. All authors have read and approved the final manuscript.

## Additional information

**Competing interests:** The authors declare no competing financial interests.

