## [Peer Review File · Nature Communications]

Reviewer #1 (Remarks to the Author):

This paper details the reaction of streamlined asymmetric α -difunctionalization of ynones. The chemistry of gold-catalyzed C-C bonds formation from ynone with electrophilic alkynylating reagents (TIPS-EBX) is now very well explored. The lead author has also performed very similar studies (for instance, ref. 33)

But the chemistry of asymmetric α -alkynylation- α -fluorination of ynones is interesting and does in my view contain material suitable for publication.

Before the publication, a major revision is necessary for the following reasons:

1. (Figure 1, D): R1, R2 is not equal
2. There are not comments on mechanism of asymmetric α -alkynylation- α -fluorination. The role of CPA and chiral amine is not clear and should be discussed.
3. The control experiment of (3a, 5c, S6e, m-xylene, -30°C) should be done.
4. "excellent ee" should be reconsidered and 90%ee is not good enough in asymmetric chemistry
5. The tolerance substituents of asymmetric α -alkynylation- α -fluorination are limited, what about the aryl and aliphatic R1 and R2 groups. The examples should be included.
6. Some symbolic errors all in this paper need to be corrected. Like:
"-40°C" should be " -40°C ", and the same symbolic errors in the paper should be corrected.
"m-xylene" "m" should be *Italic* and some common symbolic errors are existed.
"Et2O" take care the subscript
7. In addition, the coupling constants described for a number of the products do not match in pairs. There are also some strange patterns in the NMR data:
To take a single example, compound 3f, the aromatic region of the HNMR clearly has four types of hydrogen and the 5-hydrogen must be a singlet (albeit heavily tented), but the authors record what the computer says without any comment. The other hydrogens are also reported as multiplet, but clearly has a set of sharp lines for which all the coupling constants can be determined
8. The authors need to go through these recorded spectra very carefully and ensure that protons that are coupled have equal J values. The actual proton spectra need to be expanded so the reader can see the patterns of individual peaks clearly.

Reviewer #2 (Remarks to the Author):

This paper describes an enantioselective α -difunctionalization of ynones. The first step is a racemic gold-catalyzed alkynylation of enamines derived from ynones, and the second step is the enantioselective fluorination of the resulting β -alkynyl enamines using a chiral phosphate catalyst. Excellent enantioselectivity was generally obtained. The products are difficult to synthesize by currently-existing synthetic methods, and more importantly, can be transformed to other unique chiral compounds due to synthetic versatility of C-C triple bonds. Due to the high efficiency and synthetic utility of the developed method, this referee supports this paper for publication in Nature Commn. However, the following comments need to be addressed in the revised manuscript.

1. The authors previously reported the synthesis of allenes (ref. 33) under similar conditions, whereas the C-C triple bond retained under the present paper. What is key to differentiating the two pathways, to allenes or to alkynes.
2. In Table 2, the E/Z ratio varied greatly depending on the steric bulkiness of the alkyne terminal substituent. Can this variable E/Z ration be explained only from sterics? Are they kinetic products?
3. For the high enantioselectivity of α -fluorination, the conformation of C-N bond in intermediate 3a is critically important. The authors should determine the conformation and the ratio between two conformers in order to pose an image of stereodifferentiation in the following

diastereoselective fluorination. The ground state energy between conformer 3ab shown in Table 3 would not be so different from the other conformer where the C-N bond rotates and the R group goes to the down side.

4. Similarly to above point 3, enantioselectivity should also depend on the E/Z ratio of the dialkynoamine intermediate. What is the result if compound 3t with a lower E/Z ratio is used in the diastereoselective fluorination? Is the enantioselectivity dependent on the E/Z ratio of compound 3, or is there E/Z isomerization process before fluorination?

Reviewer #1 (Remarks to the Author):

This paper can be accepted, after a minor revision, as follows:

1. Some symbolic errors all in this paper need to be corrected.

Line 312-314,366 and 375 contain symbolic errors and should be corrected

2. The peaks of proton NMR are so low, and can't see clearly, please increase the level of peaks, and the expansion of H NMR should be decreased expand to -0.5-8.5

Reviewer #2 (Remarks to the Author):

The authors addressed all the points that this reviewer commented in the original submission. The revised manuscript is satisfactory. Therefore, this reviewer recommends this paper accepted in Nature Commun.

Modifications and responses are marked in red

REVIEWERS' COMMENTS:

Reviewer #1 (Remarks to the Author):

This paper can be accepted, after a minor revision, as follows:

1. Some symbolic errors all in this paper need to be corrected.

Line 312-314,366 and 375 contain symbolic errors and should be corrected.

These symbolic errors have been corrected.

2. The peaks of proton NMR are so low, and can't see clearly, please increase the level of peaks, and the expansion of H NMR should be decreased expand to 0.5-8.5.

The peaks of proton NMR have been expanded for better clarity. The expansion of HNMR has been shrunk to 0.5-8.5, with exception of several compounds that contain peaks beyond 8.5.

Reviewer #2 (Remarks to the Author):

The authors addressed all the points that this reviewer commented in the original submission. The revised manuscript is satisfactory. Therefore, this reviewer recommends this paper accepted in Nature Commun.